# Histopathology Stain-Color Normalization Using Deep Generative Models

**Farhad Ghazvinian Zanjani**    **Svitlana Zinger**    **Peter H.N. de With**
Dept. of Electrical Eng., Eindhoven University of Technology, Eindhoven, The Netherlands

**Babak E. Bejnordi**    **Jeroen AWM van der Laak**
Diagnostic Image Analysis Group, Radboud University Medical Center, Nijmegen, The Netherlands

## Abstract

Performance of designed CAD algorithms for histopathology image analysis is affected by the amount of variations in the samples such as color and intensity of stained images. Stain-color normalization is a well-studied technique for compensating such effects at the input of CAD systems. In this paper, we introduce unsupervised generative neural networks for performing stain-color normalization. For color normalization in stained hematoxylin and eosin (H&E) images, we present three methods based on three frameworks for deep generative models: variational auto-encoder (VAE), generative adversarial networks (GAN) and deep convolutional Gaussian mixture models (DCGMM). Our contribution is defining the color normalization as a learning generative model that is able to generate various color copies of the input image through a nonlinear parametric transformation. In contrast to earlier generative models proposed for stain-color normalization, our approach does not need any labels for data or any other assumptions about the H&E image content. Furthermore, our models learn a parametric transformation during training and can convert the color information of an input image to resemble any arbitrary reference image. This property is essential in time-critical CAD systems in case of changing the reference image, since our approach does not need retraining in contrast to other proposed generative models for stain-color normalization. Experiments on histopathological H&E images with high staining variations, collected from different laboratories, show that our proposed models outperform quantitatively state-of-the-art methods in the measure of color constancy with at least 10-15%, while the converted images are visually in agreement with this performance improvement.

## 1    Introduction

Computer-aided diagnosis systems are widely investigated for microscopic analysis of tissues to study presence, localization or grading of disease. Since the cells of an examined tissue lack sufficient contrast to be easily distinguished in bright-field microscopy, staining is a common practice in histopathology. Generally, two stains of contrasting colors are selected. The common laboratory stains are paired hematoxylin and eosin (H&E), which differentiate between nucleus and cytoplasm in color. In such a colorization procedure, several factors are involved that can be hard to be regularized between different medical centers or even within the samples of the same laboratory at different trials or time periods. Some sources of such variations originate from the method of specimen preparation, staining protocols like temperature of the adopted solutions, fixation characteristics, imaging device characteristics etc. [1, 2]. Such undesirable effects lead to color and intensity variations in histopathological images. Although human color perception can easily comprehend the

1st Conference on Medical Imaging with Deep Learning (MIDL 2018), Amsterdam, The Netherlands.

staining color variations among images, the design of a reliable CAD system approaching pathologists performance and robustness, is a challenging task.

Since eliminating the sources of all color variations in staining histopathology images is almost infeasible [3], two solutions have been devised. The first approach is ignoring the color information in images by transforming them into grayscale. Using texture information [4], wavelet transform [5] and local binary patterns [6] are some techniques that have been applied to grayscale histopathology images. However, bypassing the problem by eliminating the color information can loose important signal information, which is useful for medical diagnostics that are considered widely by pathologists. As a second approach, stain color normalization is broadly studied [1, 2, 3, 7, 8, 9, 10, 11, 12, 13, 14, 15, 16, 17, 18] to incorporate that signal information.

During the last decade, performance of CAD systems in computational pathology have been significantly improved, mainly by emerging new machine learning techniques like deep learning methods. Although such complex models can learn complicated segmentation or classification tasks in the presence of color variations in samples, recent studies show that stain-color normalization can lead to a higher performance in a learning system as well as a CNN model [19, 20, 1]. For example, applying stain-color normalization for the tissue classification in H&E colorectal cancer images by a CNN, remarkably improves the accuracy up to 20% [19].

## 2 Related work

The previous stain-color normalization methods can be divided into three categories: *stain-color deconvolution*, *template color matching* and *multi-task learning*, which are briefly explained below after which our contributions are described.

### 2.1 Stain-color deconvolution

Stain-color deconvolution [21] methods are considering prior knowledge of the reference stain vector for every dye and split an input RGB image into three stain channels, each representing the actual stain color. Ruifrok *et al.* [21] introduce this prior knowledge by manually selecting pixels that represent a specific stain class and then compute the color deconvolution vector. Because of the semi-automatic nature, several studies are performed later for automated extraction of stains, by e.g. using the singular value decomposition (SVD) technique [9], probabilistic Gaussian mixture model (GMM) [12], using a prior for stain matrix estimation [3] and stain-color descriptions along with training a supervised relevance vector machine [7]. Although these solutions lead to a better stain estimation, they are solely limited to image color information, while the spatial dependency between tissue structures is still ignored [1]. This aspect causes shortcomings for stain deconvolution approaches, especially when severe staining variations occur in the data.

### 2.2 Template color matching

Template color matching methods proposed by Reinhard *et al.* [8] rely on aligning the statistical color distribution (e.g. mean and standard deviation) of a *source* image with a *template* (reference) image. The authors used a set of linear transformations for assigning a unimodal distribution to each channel of the CIELAB color model. Afterwards, each channel was treated independently for alignment. Since there is a dependency between the color channels due to dye contribution, this approach has drawbacks that have been addressed in [7, 1]. For solving this problem, separate transformations are performed on stain classes [12], or on tissue classes [1, 13]. For avoiding artifacts at the border of different classes under different transformations, the authors considered a weighted contribution of these transformations in the final color image. Two proposed solutions consist of estimating weights of the GMM [12] and training a naive Bayesian classifier [1]. The latter solution introduces multiple parameters and thresholds which cannot be optimally applied to a new dataset or even fail if the tissue type changes.

### 2.3 Multi-task learning

Recently, the capacity of deep generative models has been explored for performing stain-color normalization. This involves the application of the GAN [22] in the framework of multi-task

learning (also called *stain-style transfer* model), which has been studied by some researchers [23, 24]. According to their approach, learning the color normalization task by a GAN is integrated with another *discriminative* model (e.g. supervised classification of tumors from normal tissue). The GAN model tries to convert the image color information in order to maximize the performance of a classifier. Such approach benefits from using a generative model, but defining the problem in the principle of multi-task learning has shortcomings. Firstly, this approach to color normalization is not generic and is not able to address some histology studies, e.g. when only the slide-level labels are available or in the absence of such labels. Secondly, using discriminative learning as a supervisory signal for training a GAN does not necessarily lead to an optimal color normalization, because the objective of the color normalization module differs from that of the auxiliary classifier.

In this paper, we propose using deep generative models based on CNN for fully unsupervised learning of stain-color normalization. Our contribution is twofold. 1- We present three methods, each based on a deep generative model: GAN, VAE and deep convolutional GMM. Since the classical deep generative models are not directly applicable to stain-color normalization, we have modified the standard versions of those models. 2- In contrast to the previous generative models which are used for stain-color normalization, our methods are fully unsupervised and do not need any labels or prior assumption on the H&E image contents.

## 3 Methods

In order to adapt the conventional deep generative models to stain-color normalization, we have designed our methods by considering three important properties.

1. The image structures must be preserved after color conversion and only the chromatic information can be subject to change.

2. The model does not need any labels or any assumptions about the data and should be fully unsupervised.

3. The models should learn the color transformation between any two image pairs. It means that the models do not convert the color of the input images to only a predefined reference image (or a subset of reference images). This property facilitates time-critical CAD systems in case of changing the reference image, since our approach does not need retraining but only re-adjustment of some parameters.

In the remainder of this section, these three properties are taking into account and will be guiding the explanation of our methods in detail.

### 3.1 GAN-based color normalization model

In contrast with the GAN-based solution for stain-color normalization from recent literature [23, 24], we presented recently a fully unsupervised GAN-based model for stain-color normalization [25]. The block diagram of the model is shown in Figure 1. The model was inspired by the *InfoGAN* [26] architecture, which in addition to the *generator* and *discriminator* networks, benefits from an *auxiliary* network. Here, the auxiliary network has two functions. First, it imposes a constraint for discovering a semantic structured latent variable in the generator network. Second, after training by applying the auxiliary network on any given reference image, the model adjusts the mapping parameters of the generator to produce samples which resemble the reference image. For preserving the image structure, the *lightness* channel of the input images in the CIELAB color system is supplied to the generator and only the chromatic channels are generated by the GAN. The CIELAB color space has been chosen because of its higher performance for reconstructing histopathological images [27]. For training the model, four loss functions were defined: generative, discriminative, mutual information and reconstruction losses. For more detail, we refer to [25].

### 3.2 VAE-based color normalization model

A VAE [28] consists of two networks called *encoder* and *decoder*. The encoder network maps the input data ($\mathbf{x}$) to a usually lower-dimensional latent variable ($\mathbf{z}$) and the decoder network transforms the $\mathbf{z}$ back to the original data space. In contrast to a standard auto-encoder, the VAE regularized the encoder network by imposing a prior (e.g. a unit Gaussian distribution) over the latent distribution

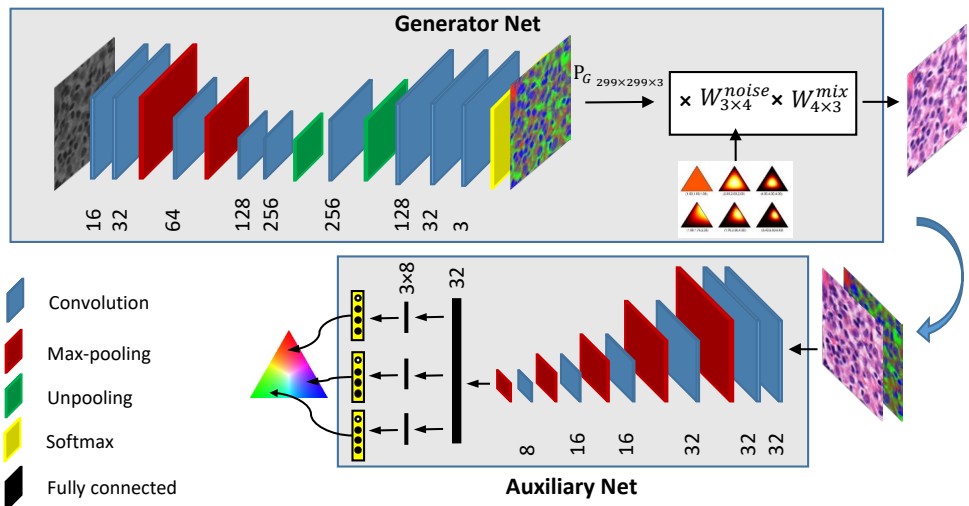

**Figure 1:** Schematic of GAN-based model [25] for stain-color normalization (the discriminator network was not shown here).

$p(\mathbf{z})$. The VAE loss includes two loss functions: generative and latent. The generative loss ($\mathcal{L}_{rec}$) is defined as the negative log-likelihood of data, which indicates how accurately the network reconstructs the input image. The latent loss ($\mathcal{L}_{latent}$) is defined by the Kullback–Leibler divergence ($D_{KL}$) that measures how closely the distribution of the latent variable $q(z|x)$ matches the prior distribution. Both losses are specified by

$$\mathcal{L}_{rec} = \mathbb{E}_{q(\mathbf{z}|\mathbf{x})}[log p(\mathbf{x}|\mathbf{z})] \quad \text{and} \quad \mathcal{L}_{latent} = D_{KL}(q(\mathbf{z}|\mathbf{x})||p(\mathbf{z})). \tag{1}$$

The VAE loss is simply the combination of these two losses, so that

$$\mathcal{L}_{VAE} = \mathcal{L}_{rec} + \mathcal{L}_{latent} = -\mathbb{E}_{q(\mathbf{z}|\mathbf{x})}[log(\frac{p(\mathbf{x}|\mathbf{z})p(\mathbf{z})}{q(\mathbf{z}|\mathbf{x})})]. \tag{2}$$

In our proposed model, the encoder input is the H&E image ($\mathbf{X} = \{\mathbf{x}_1, \mathbf{x}_2, ..., \mathbf{x}_N\}$) in the HSD [29] color system with total number of pixels (observations) equal to $N$. By applying several operations like convolutional, (un)pooling, nonlinear activation functions (ReLU) and a softmax layer at the output, the encoder transfers each pixel of input image into a $K$-dimensional vector (called $\boldsymbol{\gamma}$) in the unity range that adds up to unity ($Enc : \mathbb{R}^2 \to (0, 1)^K$). The $\boldsymbol{\gamma} = Enc(\mathbf{X})$ is interpreted as a *membership* variable, which represents the tissue-class membership of each pixel. In other words, the encoder network aims to segregate the input image into tissue structure channels. In classical VAE, the latent random variable $\mathbf{z}$ is considered as a deterministic variable which is generated at the output of the encoder, but in our proposed model the $\mathbf{z}$ denotes the color distribution of tissue classes in the input image with the mean ($\boldsymbol{\mu} = [\boldsymbol{\mu}_1, ..., \boldsymbol{\mu}_K]$) and covariance matrix ($\boldsymbol{\Sigma} = \boldsymbol{\sigma}^2\mathbf{I}$). Since such a $\boldsymbol{\mu}$ and $\boldsymbol{\Sigma}$ are conditional variables and dependent on the tissue class (latent variable $\boldsymbol{\gamma}$), consequently the $\mathbf{z}$ is treated as a latent variable. More formally, this leads to the following specification:

$$\boldsymbol{\mu}_i = \mathbb{E}(\mathbf{x}_j|argmax(\boldsymbol{\gamma}_j) = i) \quad \text{for} \quad i = 1, 2, ..., K \quad \text{and} \quad j = 1, 2, ..., N$$
$$\boldsymbol{\Sigma}_i = \mathbb{E}[(\mathbf{x}_j - \mathbb{E}[\mathbf{x}_j])(\mathbf{x}_j - \mathbb{E}[\mathbf{x}_j])^T|argmax(\boldsymbol{\gamma}_j) = i]$$
$$\mathbf{z} = \mathcal{N}(\boldsymbol{\mu}, \epsilon \cdot \boldsymbol{\Sigma})$$

Parameter $K$ stands for a predefined number of tissue classes in the images (e.g. for H&E image is set to $K = 3$ for nuclei, the surrounding tissues and the background regions). The auxiliary variable $\epsilon$ is sampled from a prior distribution (e.g. $\epsilon \sim \mathcal{N}(0, 1)$). The meaning of the above equation is that the model segments the input image into $K$ regions and computes the mean and standard deviation of the input pixels in those regions individually. For simplicity, we assume that the chromatic information between different tissue classes are uncorrelated, i.e. the covariance matrices given by $\boldsymbol{\Sigma} = \boldsymbol{\sigma}^2\mathbf{I}$ where $\mathbf{I}$ is the identity matrix, although the conclusions will hold for general covariance matrices. To summarize, the encoder is expected to segment the input images into the tissue classes (e.g. nuclei which mostly absorb hematoxylin stain, background with no stain and the other tissue structures which mainly absorb eosin stain) at its softmax layer. Afterwards, the parameters of the

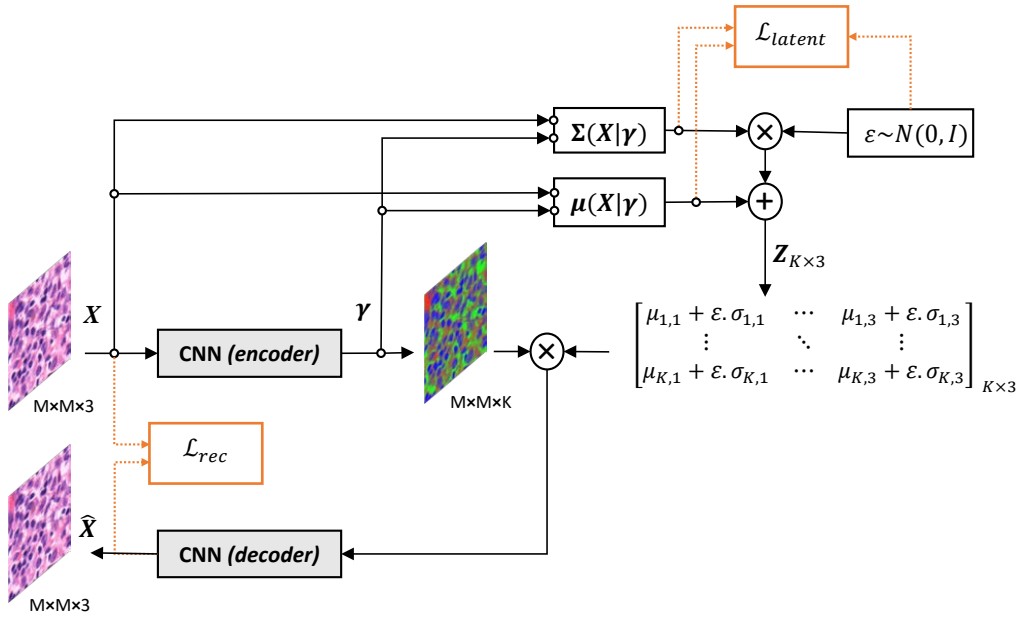

**Figure 2:** Schematic of VAE-based model for stain-color normalization (illustration has been inspired from [30]).

color distribution in the input image is computed individually per tissue class. Similar to standard VAE model, by keeping the calculated $\boldsymbol{\mu}$, we only sample from the standard deviation by a prior Gaussian distribution with zero mean and unity variance. Afterwards, the decoder network transfers the sampled latent variable into the output for reconstructing the chromatic channels of the input. Figure 2 shows the architecture of the VAE-based color normalization model.

### 3.3 Deep Gaussian mixture color normalization model

The classical GMM is a well-established machine learning technique which is applied also for stain-color normalization [12, 13]. Applying conventional GMM for stain-color normalization has some shortcomings. Because the conventional GMM-based models clusters the pixels based on their color attributes only, while it ignores the spatial information, appearance and the shape of tissue structures, its performance deteriorates, especially when strong staining variations appear in histopathological images (see Figure 3). On the other hand, several published studies that involve the spatial properties of biological structures which are occurring in most H&E images, give better results for stain-color normalization [1, 25, 31].

The recent development of deep generative models has invoked some extensions to the classical GMM [32, 33]. Two proposed approaches are (1) constructing a stack of multiple GMM layers on top of each other in a hierarchical architecture [33], and (2) using auto-encoder neural networks while applying the GMM on their low-dimensional representations [32]. Here, we introduce an alternative approach, called deep convolutional GMM (DCGMM), that represents a fully-convolutional CNN of which the parameters are jointly optimized with the parameters of the GMM, in an end-to-end learning. For further explaining this alternative method, we first detail the needed terminology for GMM.

The GMM of data ($\mathbf{x}$) can be presented as a linear superposition of $K$ Gaussian mixtures in terms of discrete *latent* ($\mathbf{z}$) variable vector, in the form of

$$p(\mathbf{x}) = \Sigma_{k=1}^{K} \pi_k \mathcal{N}(\mathbf{x}|\boldsymbol{\mu}_k, \boldsymbol{\Sigma}_k). \tag{3}$$

The $K$-dimensional binary random variable $\mathbf{z}$ has one-hot encoding ($z_k \in \{0, 1\}$; $\Sigma_{k=1}^{K} z_k = 1$), which represents tissue classes in our study. In Eq. (3), the mixing coefficient $\pi_k$ must satisfy $0 \leq \pi_k \leq 1$ together with $\Sigma_{k=1}^{K} \pi_k = 1$, in order to fulfill a valid probability definition [34]. Here, $\mathcal{N}$ stands for a multivariate normal distribution with mean $\boldsymbol{\mu}_k$ and covariance matrix $\boldsymbol{\Sigma}_k$. If we consider $\pi_k$ as prior probability of class $z_k$, its posterior probability, called *responsibility*, can be written as

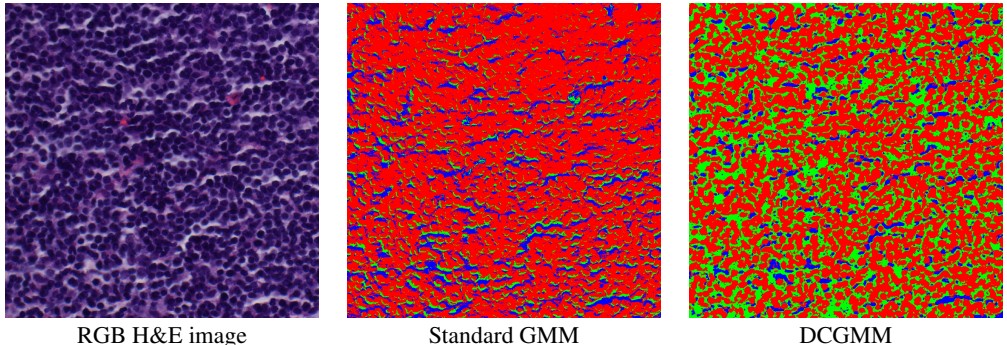

| RGB H&E image | Standard GMM | DCGMM |

**Figure 3:** Three-class tissue clustering for a standard GMM compared to the DCGMM. Red, green and blue colors represent nuclei, surrounding tissues and background regions, respectively.

follows [34]:

$$\gamma(z_k) = p(z_k = 1|\mathbf{x}) = \frac{\pi_k \mathcal{N}(\mathbf{x}|\boldsymbol{\mu}_k, \boldsymbol{\Sigma}_k)}{\Sigma_{j=1}^K \pi_j \mathcal{N}(\mathbf{x}|\boldsymbol{\mu}_j, \boldsymbol{\Sigma}_j)}. \tag{4}$$

According to Eq. (3), the (natural) log-likelihood function is

$$\ln p(\mathbf{X}|\boldsymbol{\pi}, \boldsymbol{\mu}, \boldsymbol{\Sigma}) = \Sigma_{n=1}^N \ln\{\Sigma_{k=1}^K \pi_k \mathcal{N}(\mathbf{x}|\boldsymbol{\mu}_k, \boldsymbol{\Sigma}_k)\}. \tag{5}$$

where $N$ is the total number of pixels in the input image ($\mathbf{X} = \{\mathbf{x}_1, \mathbf{x}_2, ..., \mathbf{x}_N\}$). Given the GMM, the objective is to maximize the likelihood function (Eq. (5)) with respect to the parameters ($\boldsymbol{\mu}_k$, $\boldsymbol{\Sigma}_k$, $\pi_k$). The common approach for this is using the expectation maximization (EM) algorithm by iteratively evaluating the responsibilities (Eq. (4)) and re-estimating the parameters.

The DCGMM fits a GMM to the pixel-color distribution, while being conditioned on tissue classes. For processing the image and detecting the tissue classes, the high image-representation capability of the CNN has been exploited. To do so, estimating the responsibility coefficients is performed by a CNN, similar to what the encoder network in VAE performs, as we explained earlier. Instead of using iterative EM algorithm, all parameters of the DCGMM are jointly optimized by minimizing the negative log-likelihood (maximizing Eq. (5)) with the gradient descent algorithm. The calculation of the required partial derivatives of negative log-likelihood with respect to its parameters ($\boldsymbol{\pi}, \boldsymbol{\mu}$ and $\boldsymbol{\Sigma}$) for performing a gradient descent optimization can be found in [35, p. 45]. For a better understanding, one can consider that the *E-step* in an EM-based algorithm is replaced by a CNN. Afterwards, by having a current estimate for the responsibility parameters, we can calculate the $\boldsymbol{\mu}$ and $\boldsymbol{\Sigma}$ of multivariate Gaussian distributions over the channels of an input image, similar to *M-step*. Therefore, we can compute Eq. (5) and consequently use gradient descent and the back-propagation algorithm for training the DCGMM.

After training the model, by applying the DCGMM on any given test image, the responsibility vector for each pixel in the image is computed (see Figure 4). Consequently, the parameters of multivariate Gaussian distributions are estimated. The color conversion can be performed by aligning the Gaussian distributions in a source image to its counterpart in a template image. Such a conversion consists of *shifting* the mean, *whitening* and *coloring* transformation. The coloring transformation refers to applying the SVD technique to the whitened Gaussian distribution and scale it to obtain the same covariance matrix as the template image.

## 4 Experimental Results

We focus on inter-laboratory variations of the H&E staining, as it is a major concern in large-scale application of CAD in pathology. We evaluate the performance of our three proposed generative neural networks in comparison to that of five competing state-of-the-art algorithms: linear appearance normalization by Macenko *et al.* [9], statistical color properties alignment by Reinhard *et al.* [8], nonlinear mapping for stain normalization by Khan *et al.* [7], sparse non-negative matrix factorization by Vahadane*et al.* [10] and WSI color standardization by Bejnordi *et al.* [1]. For a fair comparison between different methods, we transform the output color-converted images of our proposed methods into the RGB color coordinates.

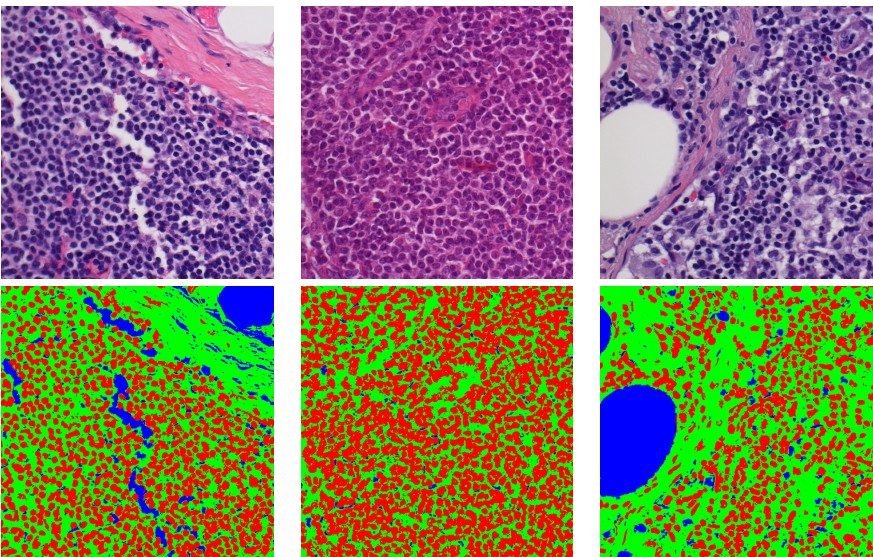

**Figure 4:** Three examples of tissue clustering by the DCGMM. (red) nuclei; (green) surrounding tissue; (blue) background.

## 4.1 Dataset

For better comparison with recent studies, we use the same dataset as has been introduced in [1]. The dataset contains 625 images (each $1388 \times 1040$ pixels) from 125 digitized H&E stained WSIs of lymph nodes from 3 patients. The tissues are sectioned and sent to five Dutch pathology laboratories, each using their own routine staining protocols (more details can be found in [1]). Since the size of histopathology WSIs are huge in dimension (e.g. in the range of $100k$ pixels in width and height), analysis of such images in the patch level is a common practice. Usually the upper bound for the patch size is defined according to the hardware memory limitations and mainly GPU memory in deep learning implementations. Our model is trained on randomly cropped patches ($299 \times 299$ pixels for VAE and GAN networks and $576 \times 576$ pixels in DCGMM experiments) and evaluated on the full-size images, since all three networks have a fully convolutional architecture. The evaluation is performed by using leave-one-out cross-validation based on the laboratories where the data was collected.

## 4.2 Results

The normalized median intensity (NMI) measure [1, 13] is used to evaluate the color constancy of the normalized images. This measure enables comparison of intensity statistics over a population of images [1]. Quantitative analysis is based on independently evaluating the color constancy in the regions that show mostly absorbed hematoxylin or eosin. Since nuclei mostly absorb hematoxylin, they first are detected automatically by using a fast radial symmetry transform and a marker-controlled watershed algorithm [1]. For evaluation of the eosin analysis, several regions are manually annotated for 25 images. The color constancy is evaluated as the standard deviation (SD) and coefficient of variation (CV) of NMI scores for the samples in different laboratories before and after color normalization using different methods. The measured SD and CV values are shown in Table 1 and illustrated in Figure 5. As we can observe from the Table 1, our proposed models show better results in eosin regions when compared with most of the previous methods. For the hematoxylin regions, our proposed models show acceptable color constancy and the DCGMM method outperforms all the state-of-the-art methods, measured as average performance on all laboratories. The VAE-based model did not show high performance in comparison with our other two deep generative models. This can originate from leaking the color information between the source image at the input of the VAE-based model to its output. In other words, the encoder could not perfectly disentangle the chromatic information from the intensity of the input image. Among our proposed methods, the DCGMM converges very rapidly after a few epochs on the training set. Among previous methods, Bejnordi *et al.* method [1] shows a higher performance and it is close to the performance of deep generative

**Table 1:** Standard Deviation (SD) and Coefficient of Variation (CV) of NMI for all five laboratories for hematoxylin and eosin dyes.

| Method | Hematoxylin | | | | | | | | | | | | Eosin | |
| --- | --- | --- | --- | --- | --- | --- | --- | --- | --- | --- | --- | --- | --- | --- |
| | Lab 1 | | Lab 2 | | Lab 3 | | Lab 4 | | Lab 5 | | Average | | | |
| | SD | CV | SD | CV | SD | CV | SD | CV | SD | CV | SD | CV | SD | CV |
| Original | 0.033 | 0.065 | 0.031 | 0.060 | 0.037 | 0.078 | 0.029 | 0.049 | 0.028 | 0.051 | 0.032 | 0.060 | 0.0563 | 0.0748 |
| Macenko[9] | 0.029 | 0.052 | 0.026 | 0.046 | 0.020 | 0.037 | 0.025 | 0.044 | 0.020 | 0.035 | 0.024 | 0.043 | 0.0362 | 0.0439 |
| Reinhard[8] | 0.032 | 0.058 | 0.025 | 0.044 | 0.020 | 0.035 | 0.030 | 0.052 | 0.029 | 0.049 | 0.027 | 0.047 | 0.0386 | 0.0494 |
| Khan[7] | 0.066 | 0.156 | 0.067 | 0.155 | 0.085 | 0.158 | 0.054 | 0.110 | 0.049 | 0.093 | 0.064 | 0.135 | 0.0434 | 0.0555 |
| Vahadane[10] | 0.036 | 0.065 | 0.032 | 0.058 | 0.024 | 0.046 | 0.023 | 0.042 | 0.020 | 0.038 | 0.027 | 0.050 | 0.034 | 0.041 |
| Bejnordi[1] | **0.016** | **0.029** | **0.015** | **0.027** | 0.018 | **0.034** | 0.029 | 0.055 | 0.024 | 0.044 | 0.021 | 0.038 | 0.0191 | 0.0220 |
| VAE | 0.029 | 0.052 | 0.022 | 0.043 | 0.021 | 0.064 | 0.028 | 0.050 | 0.025 | 0.055 | 0.025 | 0.0528 | 0.026 | 0.036 |
| GAN | 0.024 | 0.053 | 0.019 | 0.043 | 0.020 | 0.043 | 0.027 | 0.057 | 0.024 | 0.053 | 0.022 | 0.050 | 0.0195 | 0.0218 |
| DCGMM | 0.022 | 0.045 | 0.017 | 0.034 | **0.017** | 0.036 | **0.014** | **0.030** | **0.017** | **0.035** | **0.017** | **0.036** | **0.0188** | **0.0209** |

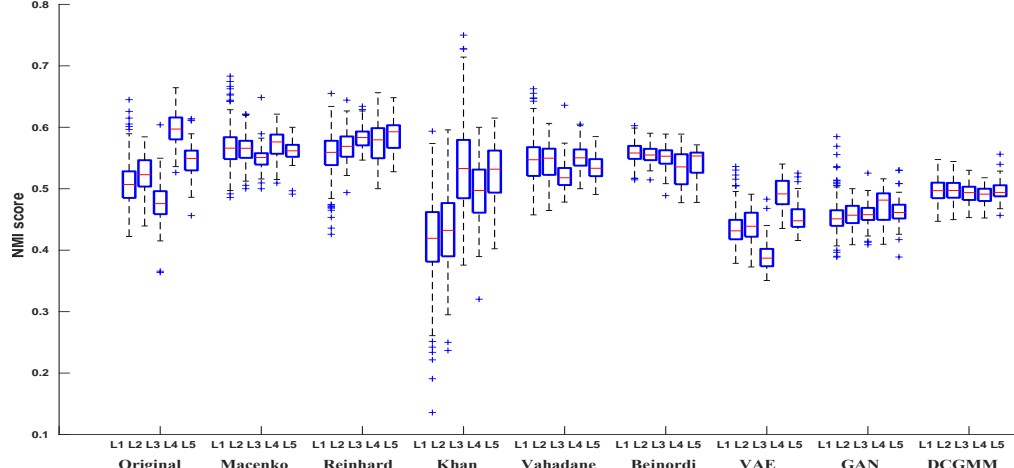

**Figure 5:** Boxplot of NMI scores in hematoxylin regions for the original images from different laboratories and their color-normalized versions by different methods.

models. However, we should consider that while our models are generic and lack any assumption about H&E image contents, their method is based on some assumption like e.g. fitting elliptical shapes to nuclei by the Hough transform and applying several thresholds within the algorithms that can be easily violated if the tissue type changes. The visual illustration of stain-color conversions between a source and reference image, resulting from the different methods, is shown in Figure 6. It can be observed that the quantitative measure has high correlation with the visual perception of the results.

# 5 Conclusion

In this paper, we have introduced three deep generative models for performing stain-color normalization in histopathological H&E images. For designing such a model, three main properties were considered: (1) the converted image should preserve the image appearance, excluding the chromatic information; (2) the model should train fully unsupervised; (3) the model should be able to convert the color information between any two unseen image pairs and not only convert the color of input images to the color of a predefined reference image. Since the classical deep generative models cannot fulfill all aforementioned considerations, we have introduced the GAN-based, VAE-based and the DCGMM methods. The experiments have shown that the DCGMM outperforms state-of-the-art methods in the measure of color constancy with at least 10-15% (or more) on a challenging dataset which contains severe staining variations between laboratories. For the near future, we aim to study the possibility of applying the generative models to a unified image dataset, consisting of more samples from different tissue types, collected from different organs.

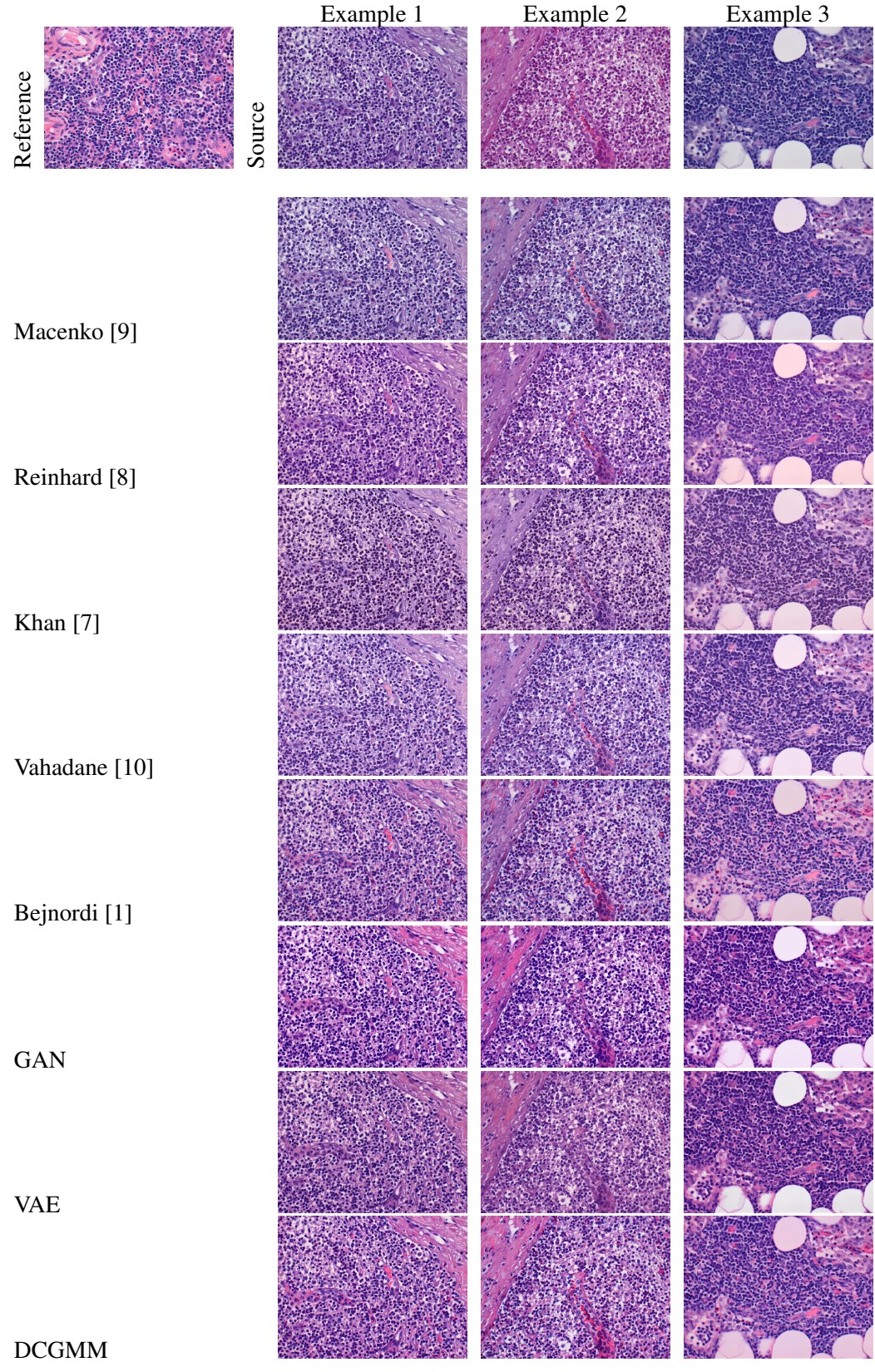

**Figure 6:** Performance of different methods on stain color-normalization between three H&E images from three different labs and a reference image.

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
