# OpenReview forum: "Histopathology Stain-Color Normalization Using Deep Generative Models"
_MIDL.amsterdam/2018/Conference — MIDL 2018 Poster_

### Review · AnonReviewer2 · 2018-05-06
**A sound paper with several approaches compared to address a topical question. The paper is however quite unclearly written thereby limiting its impact**

**Rating:** 5
**Confidence:** 2

**Review:**

The authors present three fully unsupervised methods each based on a different deep generative model for histo-pathological stain-color normalization. These three methods do not rely on any priors such as assumptions or annotations of images. While qualitatively speaking the authors’ methods seem to produce results similar to state-of-the-art, quantitative analysis based on normalized median intensity seems to indicate that the authors’ methods produce more consistent results in terms of color constancy.

Quality
The work is of good quality overall. The proposed are validated on clinical data obtained from five different national laboratories, and compared against five state-of-the-art methods, which is significant amount of work.

The paper also aims to address the important problem of constraining the proposed techniques to preserve the structures in processed images while normalizing the color. And while the latter aim seems to have been verified, the manuscript seems to lack a discussion on the former. It looks like the output of the proposed methods is aimed at being input to CAD systems. However the benefit of independent colour normalisation in comparison to end-to-end CAD approaches embedding the pre-processing should be discussed/evaluated in more depth.

Clarity
The presentation of the material is somewhat fuzzy. The three proposed methods would benefit from being presented in greater depth.

NMI was chosen for assessing the performance of the methods. Although a reference has been provided where NMI is described, the authors should also motivate this choice. What makes NMI suitable for this kind of analysis?

The authors use the CIELAB color space, however this is not motivated. It will be helpful to highlight this and briefly state why it was chosen. Also it looks like the NMI analysis was carried out on RGB images. As the different parts of the manuscript use different color spaces, this should be highlighted.

It looks like some images were manually annotated (Results, first paragraph). This should be explained in more detail such as who (clinician/engineer) annotated these and how.

Originality
The authors clearly state the novelty of their approach in the introduction. A clear quantitative and qualitative comparison with the state-of-the-art methods is provided.

Significance
The authors imply that this work is motivated by the need for a reliable CAD system. Also the need for having consistent color patterns for correct CAD seems to be a driving force in this field. However the clinical driving force could have been explained better.

Pros and cons
The quantitative results indicate that the methods outperform the state-of-the-art. However judging by a look at Fig. 5, the clinical utility of the methods is not immediately clear, as the images look quite similar to other methods. It would be interesting to have some more clinical insight on the results.


**Special Issue:**

Yes

---

### Review · AnonReviewer1 · 2018-05-08
**Clinically relevant paper with comparisons to state-of-the-art-methods. Novel contributions. Poor explanation of method limits clear understanding for adoption in the research community. Manuscript will benefit from additional information regarding the method/relevant diagrams.**

**Rating:** 4
**Confidence:** 2

**Review:**

- The authors presented 3 unsupervised methods based on GAN models for histopathological stain-color normalization. In practice, the problem is solved by employing prior information and optimizing deterministic probabilistic variables. The key aspect of the presented methods is not requiring any prior information or image annotations but learning the essential latent variables using GANs. The presented method is novel and theoretically very interesting.

-The paper can be written better and in a more organized manner. The methods require detailed explanation. For example, the paper claims it is end to end learning, but a post processing step is applied based on the variables learned=>"such a conversion consists of shifting of mean, whitening....". It is not clear how the method results in an end to end system and simplest way to address it could be adding a block diagram denoting each step of the process.

- The qualitative results seems very similar to the compared methods, but results show superior quantitative accuracy. It is not evident if the increased quantitative accuracy is clinically significant since it is not visible in the qualitative results.

-The paper may be strengthened by using a larger dataset. 625 images from 3 patients for a deep learning method is not adequate since it may be biased to limited samples. However, the authors appropriately validated using leave one out and comparing it to 5 lab annotations. This is a significant amount of validation and offsets limitation of the data used.

-What method is "original" in table 1? Is it [11]? if not why not also compare it to [11]?

-How does this method factor in or improve the CAD system? The authors should discuss this.

-Why was CIELAB color space chosen?

-"Our model is trained on randomly cropped patches (299 × 299 pixels for VAE and GAN networks and 576 × 576
pixels in DCGMM experiments)." There is not motivation or reasoning discussed behind the sizes chosen or why patch based approach was chosen in the first place.

**Special Issue:**

Yes

---

### Review · AnonReviewer3 · 2018-05-09
**The authors present a set of deep generative models for stain normalization of H&E stained histopathology images. The proposed methods are fully unsupervised and can normalize an image based on any reference image without retraining of the trained model. One of the proposed methods outperformed the existing methods in quantitative evaluation on images from five different labs.**

**Rating:** 3
**Confidence:** 2

**Review:**

Significance:
The authors highlight the importance of stain-colour normalization in the context of CAD systems in general without specifying any example of a CAD system where stain normalization is important. Specially, those ‘time-critical’ CAD systems where usage of new reference image is important.

Significance of the third property of the proposed method is not clear. A CAD system trained on images normalized based on one reference image will result in deteriorated performance on test images normalized based on another reference image.

Originality:
The authors claim that they proposed three methods in this manuscript whereas their first method has already been presented at ISBI 2018 as they mentioned in the last sentence of Section 3.1. Therefore, it is not fair to present that work as contribution in another work. However, other two methods are well modified versions of the existing methods to address the problem at hand and could be considered as novel in the histopathology domain.

Clarity:
Overall the paper has room for improvement in terms of clarity of presentation. A framework diagram of VAE-based method would be helpful for understanding the proposed method as it is not entirely clear how authors used the reference image to stain normalization of source images.

Quality:
DCGMM based method outperformed the existing methods. However, average results of VAE-based methods are not promising. In case of hematoxylin regions results are inferior to standard non-training-based methods.

Pros and cons:
Dataset for the evaluation of the proposed is quite reliable (images from five different labs). However, it is difficult to gauge the real usefulness of the proposed methods from the quantitative results. It would be best if authors evaluate the quantitative performance of their methods from training a simple network for a real CAD-related task on images normalized by different methods.


**Special Issue:**

Yes

---

### Comment · ~Bram_van_Ginneken1 · 2018-05-18
**Selection for longlist for special issue Medical Image Analysis**

Dear authors,

Congratulations on your acceptance to MIDL! We have selected your paper on the longlist for the Medical Image Analysis Special Issue. Please read this page:
https://midl.amsterdam/special-issue-in-medical-image-analysis/
Please answer the three questions that are listed on that page about your interest in submitting to the special issue, potential overlap with other publications, and related publications.

You can post your answer here directly below on openreview.net, or mail me directly at bram.vanginneken@radboudumc.nl.

Best regards, Bram

---

### Decision · Program_Chairs · 2018-05-15
**Paper61 Acceptance Decision**

Poster